# Immobilization of a Novel EST$_{BAS}$ Esterase from *Bacillus altitudinis* onto an Epoxy Resin: Characterization and Regioselective Synthesis of Chloramphenicol Palmitate

**Fengying Dong [1], Xudong Tang [1], Xiaohui Yang [1], Lin Lin [2,3], Dannong He [3], Wei Wei [1,* and Dongzhi Wei [1,*]**

[1] State Key Laboratory of Bioreactor Engineering, Newworld Institute of Biotechnology, East China University of Science and Technology, Shanghai 200237, China

[2] Shanghai Key Laboratory of Molecular Imaging, Shanghai University of Medicine and Health Sciences, Shanghai 201318, China

[3] Research Laboratory for Functional Nanomaterial, National Engineering Research Center for Nanotechnology, Shanghai 200241, China

* Correspondence: weiwei@ecust.edu.cn (W.W.); dzhwei@ecust.edu.cn (D.W.);
Tel./Fax: +86-21-64251803 (W.W.); +86-21-64252078 (D.W.)

**Abstract:** Novel gene *est$_{BAS}$* from *Bacillus altitudinis*, encoding a 216-amino acid esterase (Est$_{BAS}$) with a signal peptide (SP), was expressed in *Escherichia coli*. Est$_{BAS}$ΔSP showed the highest activity toward *p*-nitrophenyl hexanoate at 50 °C and pH 8.0 and had a half-life (T$_{1/2}$) of 6 h at 50 °C. Est$_{BAS}$ΔSP was immobilized onto a novel epoxy resin (Lx-105s) with a high loading of 96 mg/g. Fourier transform infrared (FTIR) spectroscopy showed that Est$_{BAS}$ΔSP was successfully immobilized onto Lx-105s. In addition, immobilization improved its enzymatic performance by widening the tolerable ranges of pH and temperature. The optimum temperature of immobilized Est$_{BAS}$ΔSP (Lx-Est$_{BAS}$ΔSP) was higher, 60 °C, and overall thermostability improved. T$_{1/2}$ of Lx-Est$_{BAS}$ΔSP and free Est$_{BAS}$ΔSP at 60 °C was 105 and 28 min, respectively. Lx-Est$_{BAS}$ΔSP was used as a biocatalyst to synthesize chloramphenicol palmitate by regioselective modification at the primary hydroxyl group. Conversion efficiency reached 94.7% at 0.15 M substrate concentration after 24 h. Lx-Est$_{BAS}$ΔSP was stable and could be reused for seven cycles, after which it retained over 80% of the original activity.

**Keywords:** *Bacillus altitudinis*; esterase; regioselectivity; transesterification; chloramphenicol palmitate

## 1. Introduction

Chloramphenicol was first isolated and purified from *Streptomyces venezuelae* by Burkholder in 1947 [1]. Following its discovery, chloramphenicol was synthesized via chemical methods and produced on an industrial scale, and then introduced into clinical practice as a therapeutic agent two years later [2]. Chloramphenicol was notably efficient against 95% of gram-negative bacteria and against a variety of gram-positive aerobic strains (including the dreaded strains of methicillin-resistant *Staphylococcus aureus*). However, the extensive use of chloramphenicol in the treatment of infectious diseases in humans and animals has caused increasing concerns about its effect on human health. It was subsequently proved to have severe adverse effects on humans; these effects could lead to aplastic anemia, childhood leukemia, grey baby syndrome, and bone marrow suppression [3]. Additionally, its bitter taste has led to the development of chloramphenicol alternatives. Many studies have addressed the derivatization of chloramphenicol, including selective esterification of the hydroxyl

groups at positions 1 and 3, with many studies demonstrating selective esterification of hydroxyl group 1 as the best derivatization method, which yields products such as chloramphenicol palmitate or chloramphenicol succinate esters. Chloramphenicol palmitate is an important chloramphenicol derivative and can be quickly hydrolyzed in vivo into the biologically active drug [4]. Nonetheless, the selective modification at one specific hydroxyl group in a polyhydroxy compound is a difficulty in the field of organic chemistry due to time-consuming protection and deprotection procedures to discriminate between the available hydroxyl groups. Furthermore, the separation process to remove the unwanted compounds increases costs and decreases the final yield. For this reason, biocatalytic transformations with high regioselectivity have become the standard procedure for the selective acylation of polyfunctionalized derivatives.

Among all biocatalytic processes, lipase reactions have many advantages owing to their high regioselectivity and mild incubation conditions, and maximize the efficiency of trivial processes in comparison to chemical methods [5–7]. Recently, it was reported that lipase can serve as a catalyst to regioselectively produce chloramphenicol derivatives. Ottolina et al. reported the regioselective synthesis of several derivatives using *Chromobacterium viscosum* lipase and lipase G in acetone, thus obtaining excellent yields (98%) after 24 h at 45 °C [4]. Later, Lv et al. reported the synthesis of chloramphenicol vinyl esters using Lipozyme from *Bacillus subtilis*, thereby demonstrating the versatility of hydrolases [8]. Among chloramphenicol derivatives, chloramphenicol palmitate has been studied only in a few works. Bizerra et al. obtained a yield of 98% using *Candida antarctica* lipase B at 0.25 M chloramphenicol, and Wang et al. demonstrated that an imprinted lipase nanogel (Lipozyme TL 100L, Novozymes UK Ltd, Nottingham, UK) can have a yield of 99% within 12 h at 20 °C [5,9]. However, the lipases discussed above are commercially available enzymes. To date, there have been no reports of novel lipases with high regioselectivity and high activity used to effectively synthesize chloramphenicol palmitate [10,11]. Therefore, there is an increasing demand for a kind of novel lipase from different species and with industrial applicability.

Free lipase is usually obtained in an aqueous form, giving it low stability and making its recycling difficult. Immobilization of lipase is a good way to improve its enzymatic performance. Immobilization of enzymes can increase their recyclability and improve the economic value and efficiency of their use in nonaqueous conditions [12,13]. Many immobilization methods have been reported, including encapsulation, entrapment, covalent binding, and cross-linking [14,15]. Among these methods, encapsulation and entrapment can lead to multiple interactions between the enzyme and the immobilization surface; these interactions can block the active site. However, covalent immobilization methods result in strong interactions between the enzyme and some functional groups of the immobilization surface, such as epoxy groups, amide and aldehyde, which can make enzyme leaching difficult and ensure high retention of a catalytic activity [16–18]. Among these groups, epoxy moieties are the most extensively applied for covalent binding of lipases because epoxy moieties carry simple groups for covalent binding and contain abundant active amino, phenolic, and thiol groups at a specific pH condition. Epoxy moieties have short spacer arms that are highly stable at pH 7.0 and that can support the enzyme on the surface of particles, thereby reducing steric hindrance of some immobilization enzymes and improving the whole catalytic activity [17,19–22].

In this research, we obtained a novel esterase gene ($est_{BAS}$) from *Bacillus altitudinis* and immobilized the recombinant protein without the signal peptide (SP) ($Est_{BAS}\Delta SP$) on a novel epoxy resin (Lx-105s). Furthermore, for the first time, we report Lx-$Est_{BAS}\Delta SP$ ($Est_{BAS}\Delta SP$ immobilized on Lx-105s) and free $Est_{BAS}\Delta SP$ to synthesize chloramphenicol palmitate with high regioselectivity and high conversion efficiency (94.7%, at 0.15 M substrate). In short, this novel esterase shows a potential for the industrial application of chloramphenicol palmitate synthesis.

## 2. Results and Discussion

### 2.1. Cloning of the Est$_{BAS}$ Gene and Sequence Analysis of Est$_{BAS}$

*est$_{BAS}$* and *est$_{BAS}$*ΔSP from *B. altitudinis* were amplified with primer pairs F1/R1 and F2/R1, respectively. The open reading frame of *est$_{BAS}$* was found to consist of 648 bp, encoding a protein of 216 amino acid residues with a molecular mass of 22.96 kDa and a theoretical pI of 9.85, as calculated by the ExPASy compute pI/Mw tool. The nucleotide sequence was deposited in GenBank under the accession number MH898984. Homology analysis of the amino acid sequence revealed that Est$_{BAS}$ of *B. altitudinis* shares the highest identity (99%) with the triacylglycerol lipase of *Bacillus* sp. and *B. pumilus* (GI: 639684939 and 408690817) and is 94% identical to the triacylglycerol lipase of *Bacillus* sp. LLTC93 (GI: 1360561895) (92% to *B. pumilus* AVI41987, GI: 1351330082; 89% to *B. pumilus* AFU81785 and *B. safensis*, GI: 408690815 and GI:1130455194; 77% to *B. atrophaeus* 010787545 and *B. atrophaeus* 063638677, GI:498485857 and 1027703361; and 75% identical to *B. velezensis*, GI: 818920927; Figure 1). Visual inspection of the sequence alignment uncovered conservation of amino acid residues in the regions related with catalysis and stabilization of the protein, e.g., the catalytic triad Ser111, Asp167, and His190 (Figure 1a). The cleavage site for the SP turned out to be located at residue 25, as determined by SignalP 5.0 Server (http://www.cbs.dtu.dk/services/SignalP/) [23]. Because this esterase has not been previously reported, the secondary and three-dimensional structures of Est$_{BAS}$ were predicted by means of the SWISS-MODEL server, and the structure of Est$_{BAS}$ was generated with a specific template (Protein Data Bank (PDB) code: 1T4M (54% similar and 79.01% identical); Figure 1b,c), which was viewed in PDB Viewer.

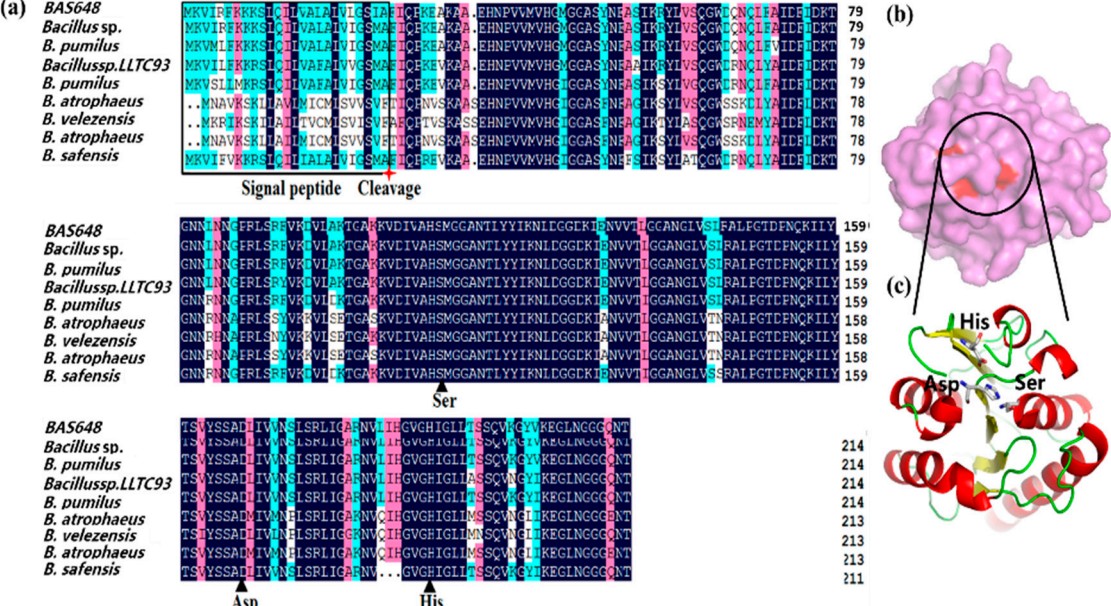

**Figure 1.** Conserved sequence alignment and structural analysis of Est$_{BAS}$. (**a**) Conserved sequence alignment of Est$_{BAS}$ and other esterase from different bacterial sources. The sequences of the following strains were obtained from GenBank: *Bacillus* sp.; *B. pumilus*; *Bacillus* sp. LLTC93; *B. pumilus* AVI41987; *B. safensis*; *B. atrophaeus* 010787545; *B. atrophaeus* 063638677; *B. velezensis*; The structures are denoted as follows: ▲, the catalytic site (Ser111, Asp167, and His190). □, the signal peptide. (**b**) The three-dimensional structure of the entire protein (Template PDB code: 1T4M, 79.01% identical). The three-dimensional structure of Est$_{BAS}$ was predicted by SWISS-MODEL server. (**c**) The catalytic site of Est$_{BAS}$ structure. The catalytic triad: Ser111, Asp167, and His190 residues are marked in gray.

## 2.2. Expression and Purification of Recombinant $Est_{BAS}\Delta SP$

Genes $est_{BAS}$ and $est_{BAS}\Delta SP$ were subcloned into vector pET-22b. The recombinant plasmid was transfected into BL21 (DE3), and the engineered strains (BL21-$Est_{BAS}$, BL21-$Est_{BAS}\Delta SP$) were grown in the Luria–Bertani (LB) medium with the ampicillin antibiotic. Optimal expression was obtained using isopropyl-β-ᴅ-thiogalactopyranoside (IPTG) at 0.1 mM and 20 °C for 20 h. Crude and purified protein are depicted in Figure 2 (purified fractions of $Est_{BAS}$ are not shown in the image of sodium dodecyl sulfate polyacrylamide gel electrophoresis; SDS-PAGE). SDS-PAGE revealed that the recombinant proteins, $Est_{BAS}$ and $Est_{BAS}\Delta SP$, have similar soluble expression profiles when induced at low temperatures (20 °C; Figure 2). $Est_{BAS}$ purification was unsuccessful: only a small amount of the purified protein was obtained (data not shown) when compared to $Est_{BAS}\Delta SP$ purified via the same Ni metal affinity chromatography (Ni-NTA) procedure. This may be because the His$_6$ tag and SP sequences were cleaved after translation, preventing the binding of $Est_{BAS}$ to the Ni-NTA agarose affinity chromatography column. The above results are consistent with those of Harris et al. [24].

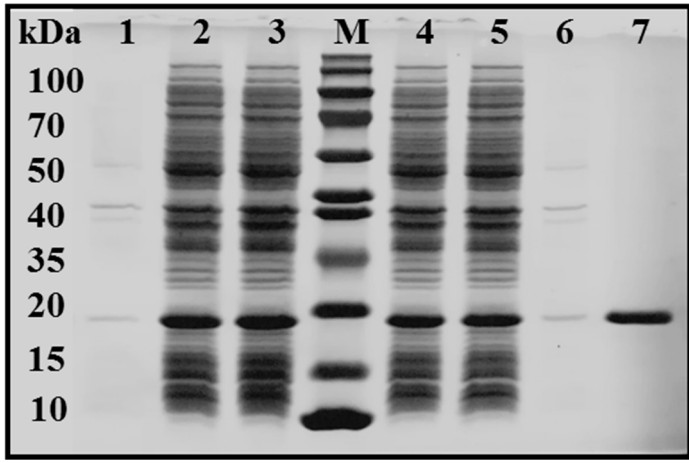

**Figure 2.** SDS-PAGE analysis of $Est_{BAS}$ and $Est_{BAS}\Delta SP$. *Lane M*: standard molecular-weight marker proteins; *lanes 3, 4*: crude extract of $Est_{BAS}$ and $Est_{BAS}\Delta SP$; *lanes 2, 5*: cell lysate supernatants ($Est_{BAS}$ and $Est_{BAS}\Delta SP$); *lanes 1, 6:* precipitates of $Est_{BAS}$ and $Est_{BAS}\Delta SP$ cell lysates; *lane 7:* purified $Est_{BAS}\Delta SP$.

## 2.3. Optimization of $Est_{BAS}\Delta SP$ Immobilization

Purified $Est_{BAS}\Delta SP$ was immobilized onto novel epoxy resin Lx-105s at 30 °C. The immobilization process is outlined in Scheme 1a. The effects of the initial amount of esterase, immobilization time on the loading capacity, and the relative esterase activity were investigated. The relative activity and $Est_{BAS}\Delta SP$ loading gradually increased to a peak (78.5 mg/g) when 90 mg of esterase was initially added (Figure 3a). After that, the relative activity decreased even if $Est_{BAS}\Delta SP$ loading was slightly augmented. Considering both esterification activity and esterase loading, 90 mg esterase was selected for immobilization to the epoxy resin Lx-105s. The above results may be explained in that unsaturated carriers provide abundant sites for esterase to immobilize to, hence improving the relative activity; whereas when more esterase was added, it caused the overcrowded accumulation of esterase onto the epoxy resin, leading to some active sites to be blocked [25]. Silva et al. obtained the same results when immobilizing lipases derived from *Candida rugosa* onto dialdehyde polyethylene glycol (PEG)-modified particles with physical adsorption style. They found that the open conformation of the enzyme was necessary to keep lipase activity high and to keep the enzyme stable [26]. As shown in Figure 3b, the loading capacity was time-dependent and the optimum time for esterase immobilization was 7 h with the loading efficiency reaching approximately 96 mg/g. After an increase in duration from 7 to 14 h, the relative activity decreased gradually and loading tended to be stable. This may be because the esterase was denatured with long-term shaking, causing a further decrease in activity at 30 °C [27].

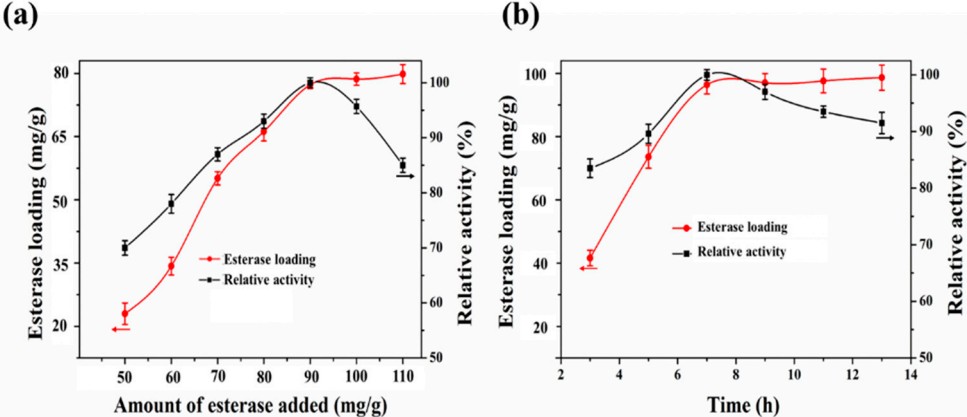

**Scheme 1.** (**a**) Est$_{BAS}$ΔSP immobilization on epoxy resin Lx-105s and (**b**) the application of immobilized Est$_{BAS}$ΔSP in the regioselective synthesis of chloramphenicol palmitate.

**Figure 3.** (**a**) Effects of the initial esterase concentration on esterase loading and relative activity. (**b**) The influence of the immobilization time on esterase loading and relative activity.

## 2.4. Characterization of Lx-Est$_{BAS}$ΔSP by FTIR

FTIR was used to investigate the potential interactions between the amino groups of the enzyme and the epoxy groups of the Lx-105s resin (Figure 4). Figure 4 shows FTIR spectra of the Lx-105s resin (a), Lx-Est$_{BAS}$ΔSP (b), and free Est$_{BAS}$ΔSP (c). The free Lx-105s resin had characteristic peaks centered at 3063 and 908 cm$^{-1}$, which are commonly assumed to be the epoxy groups stretching bands. Lx-Est$_{BAS}$ΔSP had corresponding characteristic peaks at 3060 and 908 cm$^{-1}$, but these were not present in the spectra of the free Est$_{BAS}$ΔSP indicating the presence of epoxy groups in Lx-Est$_{BAS}$ΔSP. Furthermore, absorbance bands at 3189 and 1631 cm$^{-1}$ for the amide groups of the free lipase were present, while for Lx-Est$_{BAS}$ΔSP the amide group bands shifted from 1631 to 1636 cm$^{-1}$. Additionally, the broad peak appeared in a range of 3100–3600 cm$^{-1}$ corresponding to hydroxyl and amino groups stretching. These results indicated that Est$_{BAS}$ΔSP was successfully immobilized onto Lx-105s resin [28,29].

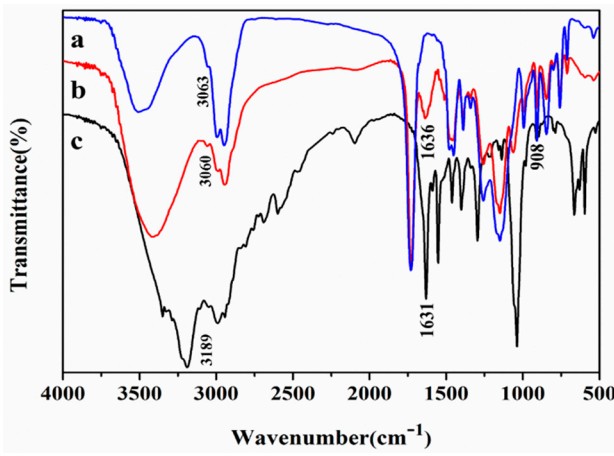

**Figure 4.** FTIR spectra of free Est$_{BAS}$ΔSP, Lx-Est$_{BAS}$ΔSP, and free epoxy resin Lx-105s. a, resin Lx-105s; b, Lx-Est$_{BAS}$ΔSP; c, free Est$_{BAS}$ΔSP.

## 2.5. Substrate Specificity of Est$_{BAS}$ΔSP

The substrate specificity of purified Est$_{BAS}$ΔSP was determined using *p*-NP esters with variable chain length: *p*-NP acetate (*p*NPA), *p*-NP butyrate (*p*NPB), *p*-NP hexanoate (*p*NPH), *p*-NP caprylate (*p*NPC), *p*-NP decanoate (*p*NPD), *p*-NP laurate (*p*NPL), *p*-NP myristate (*p*NPM), and *p*-NP palmitate (*p*NPP) at 60 °C and pH 8.0. The substrate specificity of Est$_{BAS}$ΔSP towards the *p*-NP esters of various fatty acids is presented in Figure 5. The results showed that Est$_{BAS}$ΔSP hydrolyzed *p*NPP (acyl chains of up to 16 carbons). Est$_{BAS}$ΔSP manifested higher activity for short-chain fatty acids (C < 10) compared with long-chain fatty acids (C ≥ 10; Figure 5). The highest specific activity toward *p*NPH was 1414 U mg$^{-1}$. Taken together, these results indicated that Est$_{BAS}$ΔSP is an esterase that preferentially hydrolyzes short acyl chain (C < 10) substrates [30] and has a broad spectrum of substrates: from *p*NPA to *p*NPP.

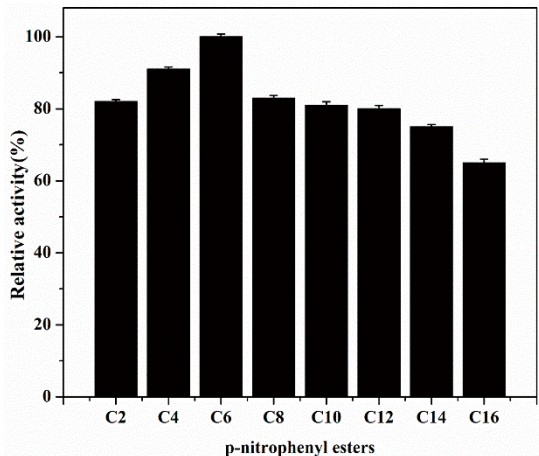

**Figure 5.** Determination of the optimum substrate for recombinant Est$_{BAS}$ΔSP. The relative activity (%) was calculated by comparison of the data between substrates and *p*-NP hexanoate (*p*NPH).

## 2.6. Effects of pH and Temperature on the Activity of Free Est$_{BAS}$ΔSP and Lx-Est$_{BAS}$ΔSP

Data on pH of a reaction help to understand the structure–function relationship of an enzyme [31]. Therefore, the impact of pH on the enzymatic properties of Lx-Est$_{BAS}$ΔSP and free Est$_{BAS}$ΔSP was investigated at various pH levels ranging from 4.0 to 10.0 with *p*NPH as a test substrate at 60 °C (Figure 6a). This figure shows that free Est$_{BAS}$ΔSP retained a maximal activity above 60% at pH from 6.0 to 9.0 with an optimal pH of 8.0. The activity significantly decreased at pH below 6.0, with approximately 30% of the maximal activity observed at pH 4.0, meaning that free Est$_{BAS}$ΔSP is

an alkaline esterase. Nevertheless, after immobilization, Est$_{BAS}$ΔSP retained approximately 80% of activity at pH from 6.0 to 9.0, and >50% of the maximal activity was observed at values below pH 8.0. This finding suggested that Lx-Est$_{BAS}$ΔSP could endure a wider pH range. The possible reason is that immobilization maintains the enzyme in a more stable conformation [32,33].

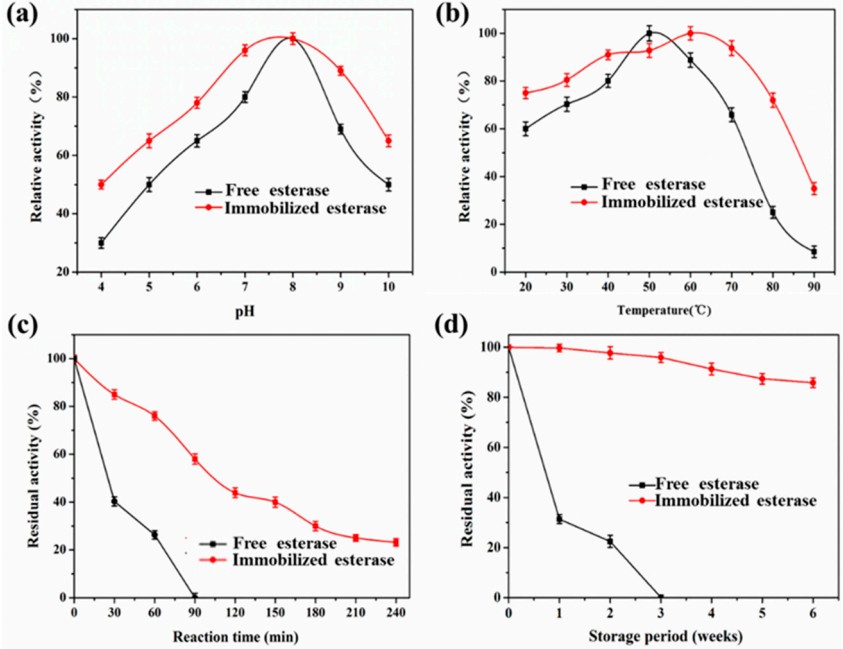

**Figure 6.** Effects of (**a**) pH and (**b**) temperature on the activity of Lx-Est$_{BAS}$ΔSP and free Est$_{BAS}$ΔSP, and the influence of (**c**) temperature at 60 °C and (**d**) storage duration (at 4 °C) on the stability of Lx-Est$_{BAS}$ΔSP and free Est$_{BAS}$ΔSP.

The effects of temperature on the activity of Lx-Est$_{BAS}$ΔSP and free Est$_{BAS}$ΔSP were investigated at pH 8.0 with *p*NPH as the test substrate (Figure 6b). Free Est$_{BAS}$ΔSP was found to have a relatively narrow range (40–60 °C) with an optimal temperature of 50 °C and retained only 60% of its maximal activity at 20 °C. Thermal stability of Lx-Est$_{BAS}$ΔSP was much higher as compared with free Est$_{BAS}$ΔSP. Lx-Est$_{BAS}$ΔSP manifested broad temperature stability, ranging from 30 to 70 °C, and the optimal temperature of Lx-Est$_{BAS}$ΔSP was found to be 60 °C. Additionally, Lx-Est$_{BAS}$ΔSP is capable of retaining 75% of its maximal activity at 20 °C and 35% at 90 °C. In contrast, the activity of free Est$_{BAS}$ΔSP decreased to 2% at 90 °C. Similar results were obtained by Li et al., who demonstrated that the activity of an immobilized lipase on an epoxy resin (ECR8285) increases by 40% when compared with free lipase at 50 °C [29]. The increased temperature stability of Lx-Est$_{BAS}$ΔSP may be attributed to the rigid structure of Est$_{BAS}$ΔSP; this rigidity resulted in a wider temperature endurance range.

## 2.7. Effects of Metal Ions and Detergents on Lx-Est$_{BAS}$ΔSP and Free Est$_{BAS}$ΔSP

The influence of some chemicals on Lx-Est$_{BAS}$ΔSP and free Est$_{BAS}$ΔSP was investigated (Table 1). Table 1 suggests that the activity of Lx-Est$_{BAS}$ΔSP and free Est$_{BAS}$ΔSP is inhibited by most of the heavy metal ions. Lx-Est$_{BAS}$ΔSP and free Est$_{BAS}$ΔSP were highly active (>90%) at low metal ion concentrations (1 mM). In contrast, $Mg^{2+}$ slightly activated Est$_{BAS}$ΔSP (105%, 108%) at low concentrations (1 mM). Nevertheless, the activity of Lx-Est$_{BAS}$ΔSP and free Est$_{BAS}$ΔSP decreased quickly at the concentrations ranging from 5 to 10 mM. Of note, Lx-Est$_{BAS}$ΔSP showed a higher activity than free Est$_{BAS}$ΔSP at concentrations ranging from 1 to 10 mM. Similarly, Mishra et al. also observed this activation function by $Mg^{2+}$. He observed that Lecitase® Ultra phospholipase activity had a 1.5-fold activation in the presence of 10 mM $Mg^{2+}$. He explained that salts are well known to cause changes in enzyme activities

via different mechanisms, such as $Mg^{2+}$, which can activate the enzyme to a lesser extent, and $Mg^{2+}$ also can function without other metal ions [34].

The detergents had various effects on the activity of esterase (Table 2). As presented in Table 2, a slight decrease in the activity of esterase (94%) was observed upon the addition of 1% of Tween 20, Tween 60 (94.5%), or Tween 80 (94.8%) compared with the control. In contrast, Lx-Est$_{BAS}\Delta$SP had a higher activity (95%, 96%, and 95%, respectively). On the contrary, the activity of Lx-Est$_{BAS}\Delta$SP and free Est$_{BAS}\Delta$SP was significantly inhibited by 1% Triton X-100 (to 38% and 36.1%, respectively) and SDS (to 20% and 18.9%). Furthermore, the activity of free Est$_{BAS}\Delta$SP disappeared at SDS and Triton X-100 concentrations up to 10%, accordingly, activity of Lx-Est$_{BAS}\Delta$SP was 0% and 8%. As a result, Lx-Est$_{BAS}\Delta$SP manifested enhanced resistance to high concentrations of metal ions and detergents. The reason was that the formed covalent bond with the support limited the conformational changes of Est$_{BAS}\Delta$SP, resulting in a more rigid structure of Lx-Est$_{BAS}\Delta$SP. However, the loss of enzymatic activity was due to some specific interaction between detergents and the enzyme surface [34].

**Table 1.** Effects of metal ions on free Est$_{BAS}\Delta$SP and Lx-Est$_{BAS}\Delta$SP activity.

| Ion | Residual | | Activity (%) [a] | | | |
| --- | --- | --- | --- | --- | --- | --- |
| | (1 mM) | | (5 mM) | | (10 mM) | |
| | F [b] | L [c] | F [b] | L [c] | F [b] | L [c] |
| None | 100 ± 0.0 | 100 ± 0.0 | 100 ± 0.0 | 100 ± 0.0 | 100 ± 0.0 | 100 ± 0.0 |
| $Mg^{2+}$ | 105 ± 2.1 | 108 ± 1.9 | 103 ± 1.9 | 104 ± 2.1 | 88.2 ± 1.9 | 99.8 ± 1.7 |
| $Zn^{2+}$ | 96.1 ± 3.7 | 97 ± 2.0 | 85.1 ± 2.7 | 87 ± 2.0 | 68.5 ± 3.5 | 70 ± 1.2 |
| $Co^{2+}$ | 92.7 ± 2.9 | 100 ± 1.7 | 81.8 ± 3.5 | 96 ± 2.5 | 71.4 ± 2.4 | 74 ± 1.5 |
| $Ni^{2+}$ | 91.5 ± 2.3 | 92 ± 2.5 | 90.6 ± 2.4 | 91 ± 2.3 | 83.2 ± 2.0 | 87 ± 2.2 |
| $Fe^{2+}$ | 92.8 ± 3.4 | 93 ± 1.8 | 83.4 ± 2.6 | 84 ± 1.7 | 68.0 ± 1.7 | 76 ± 1.6 |

[a] Lx-Est$_{BAS}\Delta$SP and free Est$_{BAS}\Delta$SP were pre-incubated with the metal ions for 1 h at 4 °C. The presented results are the mean ± standard deviation. The activity toward *p*NPH without any metal ions was set to 100%. [b] F: Free Est$_{BAS}\Delta$SP, [c] L: Lx-Est$_{BAS}\Delta$SP.

**Table 2.** The impact of detergents on the activity of Lx-Est$_{BAS}\Delta$SP and free Est$_{BAS}\Delta$SP.

| Detergent | Residual | | Activity (%) [a] | |
| --- | --- | --- | --- | --- |
| | 1% | | 10% | |
| | F [b] | L [c] | F [b] | L [c] |
| None | 100 ± 0.0 | 100 ± 0.0 | 100 ± 0.0 | 100 ± 0.0 |
| Tween20 | 94.0 ± 2.1 | 95 ± 1.9 | 40.5 ± 1.9 | 41 ± 1.1 |
| Tween60 | 94.5 ± 1.6 | 96 ± 2.1 | 45.6 ± 1.8 | 47 ± 1.9 |
| Tween80 | 94.8 ± 1.8 | 95 ± 2.3 | 53.3 ± 2.1 | 55 ± 2.1 |
| SDS | 18.9 ± 1.2 | 20 ± 1.7 | 0 | 0 |
| TritonX-100 | 36.1 ± 3.7 | 38 ± 2.9 | ND [d] | 8 ± 0.9 |

[a] The assay was performed under optimum conditions. The residual activity was measured in Tris-HCl buffer (50 mM, pH 8.0) at 60 °C with *p*NPH as the substrate. Values represent the mean ± standard deviation (*n* = 3). The activity measured without additives was set to 100%. [b] F: Free Est$_{BAS}\Delta$SP. [c] L: Lx-Est$_{BAS}\Delta$SP. [d] ND: not detectable.

## 2.8. Stability of Lx-Est$_{BAS}\Delta$SP and Free Est$_{BAS}\Delta$SP

Thermal stability of Lx-Est$_{BAS}\Delta$SP and free Est$_{BAS}\Delta$SP is described in Figure 6c. The activity of Lx-Est$_{BAS}\Delta$SP decreased gradually with incubation time, whereas the activity of free Est$_{BAS}\Delta$SP decreased almost linearly during 90 min of incubation. Furthermore, Lx-Est$_{BAS}\Delta$SP retained approximately 60% of its activity at 60 °C after 90 min of incubation, whereas free Est$_{BAS}\Delta$SP lost almost all activity under the same conditions. This was likely due to the stability of the protein three-dimensional structure of the immobilized enzyme. Xie et al. reported that an immobilized enzyme has lower sensitivity to temperature and that the conformational integrity of the enzyme structure changes due to its covalent binding to the carrier when compared with the free enzyme [35,36].

Storage stability is another important factor for industrial applications of immobilized enzymes. A comparison of storage stability between free and immobilized enzymes is illustrated in Figure 6d. Free and immobilized enzymes followed a similar trend, but the decrease in the Lx-Est$_{BAS}$ΔSP activity was lesser and slower than that of the free Est$_{BAS}$ΔSP. Lx-Est$_{BAS}$ΔSP retained over 85% of its initial catalytic activity after storage for six weeks (4 °C), whereas the free enzyme lost ~80% of its activity after only two weeks and lost almost all its activity in less than three weeks. These results clearly indicated that enzyme immobilization significantly increased storage stability, in line with another report [37].

In organic reaction systems, the organic-solvent tolerance of enzymes is especially important. Here, four organic solvents (1, 4-dioxane, ethanol, acetone, and acetonitrile) were tested (Table 3). The activity of both Lx-Est$_{BAS}$ΔSP and free Est$_{BAS}$ΔSP decreased after 70 h of incubation. For free Est$_{BAS}$ΔSP, incubation with 90% (*v/v*) 1, 4-dioxane, ethanol, and acetone reduced the activity to 12.6%, 52.5%, and 25.0%, respectively, with undetectable activity in the presence of acetonitrile. Lx-Est$_{BAS}$ΔSP retained 80.2%, 92.5%, 93.8%, and 92% of its activity in the presence of 1, 4-dioxane, ethanol, acetone, and acetonitrile, respectively, revealing that Lx-Est$_{BAS}$ΔSP has excellent organic-solvent tolerance. These results are consistent with other research [38]. Li et al. have reported that organic solvents strongly interact with enzymatic active sites; therefore, their presence could quickly render the enzyme inactive. However, the presence of the carrier may change the enzyme structure in terms of protein folding or prevent the direct interaction of the organic solvent with the enzyme [38].

**Table 3.** Effects of organic solvents on free Est$_{BAS}$ΔSP and Lx-Est$_{BAS}$ΔSP activity.

| Organic Solvents | Log*P* [a] | Residual Activity (%) [b] | |
|---|---|---|---|
| | | F [b] | L [c] |
| Control | | 100 ± 1.2 | 100 ± 0.9 |
| 1, 4-dioxane | −1.0 | 12.6 ± 0.7 | 80.2 ± 2.9 |
| Ethanol | −0.24 | 52.5 ± 1.8 | 92.5 ± 1.6 |
| Acetone | −0.23 | 25.0 ± 1.1 | 93.8 ± 2.6 |
| Acetonitrile | −0.15 | ND [d] | 92.0 ± 1.5 |

[a] The log*P* value is the partition coefficient of an organic solvent between water and *n*-octanol phases. After incubation of EST$_{BAS}$ΔSP for 70 h in one of four organic solvents, the residual activity was measured in Tris-HCl buffer (50 mM, pH 8.0) at 60 °C with *p*NPH as the test substrate. An enzyme sample incubated in the buffer alone served as a control (100% activity). [b] F: Free Est$_{BAS}$ΔSP, [c] L: Lx-Est$_{BAS}$ΔSP, [d] ND: not detectable.

### 2.9. Regioselective Synthesis of Chloramphenicol Esters by Lx-Est$_{BAS}$ΔSP and Free Est$_{BAS}$ΔSP

To assess the catalytic activity of the immobilized enzyme and free enzyme, we tested them as catalysts in the regioselective synthesis of chloramphenicol esters in acetone at 60 °C (Scheme 1b). As illustrated in Figure 7a, free enzyme had a slightly faster conversion rate than immobilized enzyme in the first three hours, then the immobilized enzyme maintained a similar conversion trend with the free enzyme from 6 to 24 h. From 21 to 24 h, the conversion rate was stable, reaching 94.7% and 90% for Lx-Est$_{BAS}$ΔSP and free Est$_{BAS}$ΔSP, respectively, at 0.15 M chloramphenicol (Figure 7a). The free enzyme featured faster conversion velocity, especially in the initial three hours; this property may be attributed to improved mass transfer in the procedure of esterification. On the other hand, Lx-Est$_{BAS}$ΔSP manifested the highest catalytic efficiency, which could be due to its good catalytic activity, organic-solvent tolerance, and thermal stability. Therefore, the enzyme could be in contact with the substrate for a sufficiently long time [29].

Reusability of the catalyst plays a significant role in potential large-scale industrial applications. Lx-Est$_{BAS}$ΔSP retained excellent residual activity (80%) after seven cycles (Figure 7b); Li et al. also observed that free lipase immobilized on epoxy resin retains 98% of its initial activity after seven cycles [29]. The above results proved that the enzyme bound to the epoxy resin via a covalent linkage has the advantage of functional groups of the support matrix and strong bonds, which ensure the high retention of catalytic activities. However, a decrease in the enzymatic activity was caused by stripping

of the bound water molecules from the enzyme by hydrophilic organic solvents in the reaction system, and the increased hydrophilic-solvent content resulted in a reduction in enzymatic activity [37].

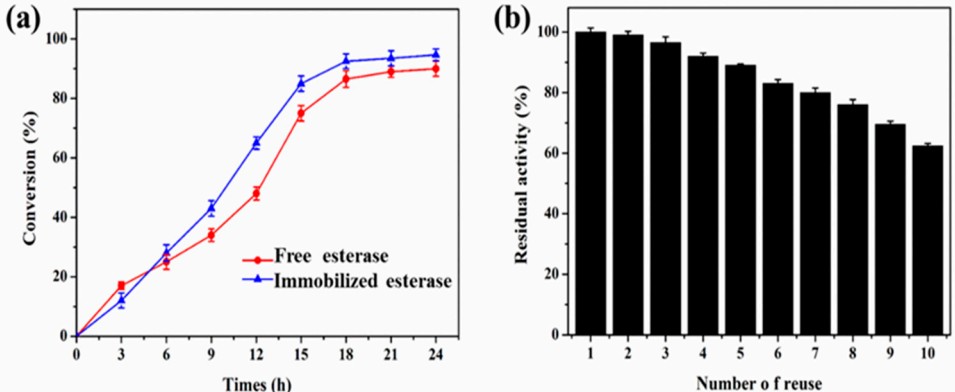

**Figure 7.** (**a**) Conversion of chloramphenicol and vinyl palmitate to chloramphenicol palmitate catalyzed by Lx-Est$_{BAS}$ΔSP and free Est$_{BAS}$ΔSP at 60 °C; (**b**) reusability of Lx-Est$_{BAS}$ΔSP.

## 2.10. Product Identification as Chloramphenicol Palmitate

The synthesis of chloramphenicol palmitate from vinyl palmitate and chloramphenicol was catalyzed by Lx-Est$_{BAS}$ΔSP or free Est$_{BAS}$ΔSP in acetone. The products were purified by thin-layer chromatography (TLC) and silica gel chromatography, and characterized by FTIR analysis (Figure 8). FTIR analysis is depicted in Figure 8. FTIR spectra featured peaks for the C=O stretching of the ester at 1742 cm$^{-1}$ and for the amide group, N–H, at 3323 cm$^{-1}$. The peak at 3504 cm$^{-1}$ was assigned to the hydroxyl group, and the peaks at 2918 and 2850 cm$^{-1}$ were assigned to stretching vibrations of CH, respectively. These findings confirmed the synthesis of the chloramphenicol esters [39].

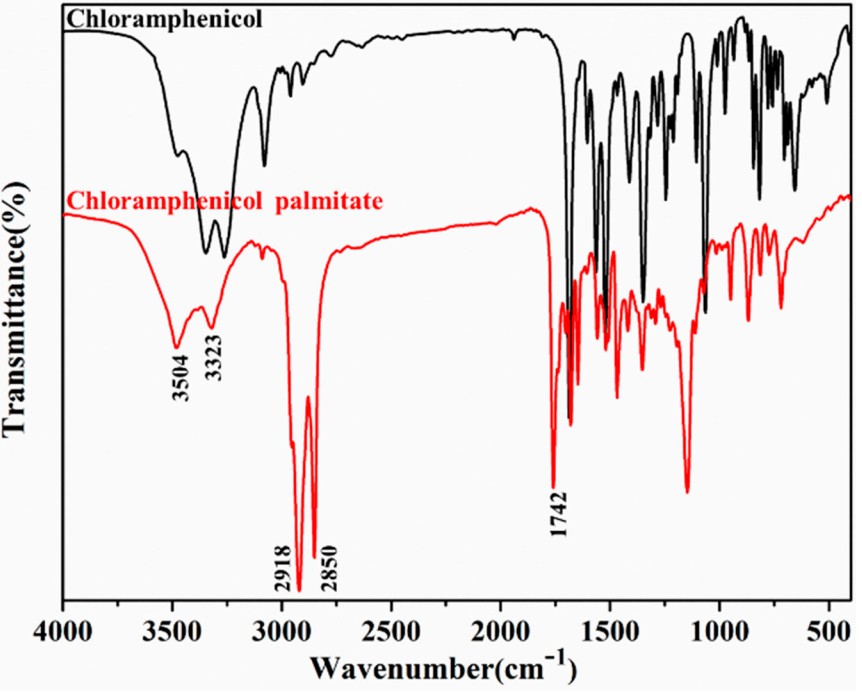

**Figure 8.** FTIR spectra of chloramphenicol (black) and chloramphenicol palmitate (red).

## 3. Materials and Methods

### 3.1. Materials

Epoxy resin Lx-105s was provided by Xian Lanxiao Technology New Materials Co., Ltd. (Xi-an, China). *B. altitudinis* was maintained in our laboratory. PrimeSTAR HS DNA Polymerase, DNA marker, and restriction enzymes were purchased from TaKaRa Biotechnology Corporation (Otsu, Japan), the ClonExpress® II One Step Cloning Kit from Vazyme (Vazyme Biotech, Nanjing, China), and the BSA standard solution from TIANGEN (TIANGEN Biotech, Beijing, China). IPTG and ampicillin were acquired from Amresco (Shanghai Genebase Co., Ltd., Shanghai, China), whereas the DNA Mini Kit and Plasmid Mini Prepare Kit were from Axygen Biosciences (Union City, CA, USA). *E. coli* DH5α (Invitrogen, USA) and plasmid pMD19-T (TaKaRa, Shiga, Japan) were employed for gene cloning and sequencing. Plasmid pET-22b (Novagen, Madison, WI, USA) served as a vector to construct the protein expression plasmid for *E. coli* BL21 (DE3). The *p*-NP ester substrates were bought from Sigma-Aldrich (St. Louis, MO, USA), whereas the yeast extract and tryptone were from Shanghai Sangon Biotechnology Co., Ltd. (Shanghai, China). All other analytical chemicals were purchased at local markets. The substrates, *p*-NP esters, chloramphenicol, and vinyl palmitate were bought from Sigma-Aldrich (St. Louis, MO, USA). Acetone and other chemicals were of analytical reagent grade and were used without further purification.

### 3.2. Triacylglycerol Esterase and Gene Analysis

The nucleotide and predicted amino acid sequences were analyzed by means of the Basic Local Alignment Search Tool (BLAST) software (National Center for Biotechnology Information: NCBI, Bethesda, MD, USA). The primers for triacylglycerol esterase gene cloning with and without the SP were designed based on the information obtained from http://www.cbs.dtu.dk/services/SignalP/. The three-dimensional protein model (http://swissmodel.expasy.org/SWISS-MODEL.html) was visualized and analyzed in PDB viewer, and the final figure was constructed in PyMol.

### 3.3. Cloning of the Esterase Gene from B. altitudinis

To determine the role of the SP in Est$_{BAS}$ purification, two plasmids were constructed. The full-length *est*$_{BAS}$ gene and *est*$_{BAS}$ without the SP (*est*$_{BAS}$ΔSP) were cloned using primer pairs F1/R1 and F2/R1, respectively (forward primer F1, 5'-TAAGAAGGAGATATACATATGATGAAAGT GATTCGATTTAAGAAA-3'; reverse primer R1, 5'-GTGGTGGTGGTGGTGCTCGAGATTCGTAT TCTGTCCTCCGCCATTC-3'; forward primer F2, 5'-TAAGAAGGAGATATACATATGTTCATCCA GCCGAAAGAAGCCA-3'; the underlined nucleotides and restriction sites). The amplification was carried out as follows: initial denaturation at 94 °C for 4 min; then 35 cycles of 94 °C for 1 min, 55 °C for 1 min, and 72 °C for 1 min; and a final extension of 10 min at 72 °C. The polymerase chain reaction (PCR) products were cloned into the pMD-19T (simple) vector after purification with the DNA Gel Extraction Kit (Axygen, Union City, CA, USA) and then transferred into *E. coli* DH5α. The target gene was obtained and sequenced. After digestion with *Nde* I and *Xho* I, the PCR products were recovered and ligated into the pET-22b vector. Recombinant plasmids were transfected into *E. coli* BL21 (DE3) cells. The recombinant triacylglycerol esterase strains (named BL21-Est$_{BAS}$, BL21-Est$_{BAS}$ΔSP) were maintained for further analysis.

### 3.4. Effects of the SP on Overexpression and Purification of Triacylglycerol Esterase

Cells containing the recombinant plasmids were incubated at 37 °C in 50 mL of the LB medium containing ampicillin (50 µg mL$^{-1}$). Cultures were induced at optical density at 600 nm (OD$_{600}$) of 0.6–0.8 by the addition of 0.1 mM IPTG and were incubated for additional 20 h at 20 °C. The effects of the SP on overexpression and purification were investigated by SDS-PAGE. The strains were harvested by centrifugation, resuspended in lysis buffer (NPI 10: 50 mM NaH$_2$PO$_4$, 300 mM NaCl, and 10 mM imidazole, pH 8.0) and sonicated, and the crude enzyme was collected by centrifugation (12,000× *g*,

10 min). The crude enzyme was passed through a 0.22 μm filter to remove any large impurities. This solution was then loaded onto a Ni-NTA Superflow Column (1 mL, Qiagen, Hilden, Germany) equilibrated with lysis buffer. The enzyme was eluted with a linear gradient of imidazole (from NPI 50 to NPI 250). The crude and purified proteins were examined by SDS-PAGE. All purification steps were carried out at 4 °C.

### 3.5. Immobilization of Purified Est$_{BAS}$ΔSP

The Est$_{BAS}$ΔSP suspension was desalted and concentrated by ultrafiltration in a 50 mL Amicon Ultra Centrifugal Filter Device with a molecular weight cutoff of 10 kDa (Millipore, Billerica, MA, USA). The hydrolytic activity of Est$_{BAS}$ΔSP was 396 ± 10.5 U/mL, as determined by the method of Wang et al. [40]. The immobilization procedure was as follows: 20 g of the Lx-105s epoxy resin was equilibrated with 100 mM phosphate buffer (pH 8.0) in a 200 mL conical flask and stirred at 30 °C for 1 h (200 rpm). The mixture was then dried by vacuum filtration. After that, 1 g of a wet carrier and different amounts of purified Est$_{BAS}$ΔSP mixed with 10 mL of a phosphate buffer were added into a conical flask (200 mL) and stirred in a thermostatic air bath shaker for 7 h (30 °C, 200 rpm). After that, the suspension was filtered, and the Lx-Est$_{BAS}$ΔSP was washed with the buffer until no protein was detected in the buffer. At last, Lx-Est$_{BAS}$ΔSP was dried (25 °C, 10 h) in a vacuum desiccator and stored (4 °C) until experiments.

### 3.6. Biochemical Characterization of Lx-Est$_{BAS}$ΔSP and Free Est$_{BAS}$ΔSP

The enzymatic activity of purified Est$_{BAS}$ΔSP in solution was assayed by measuring the absorbance of liberated *p*-NP at 405 nm. One unit of enzymatic activity was defined as the amount of enzyme needed to release 1 μmol of *p*-NP per minute [37].

Reactions were monitored at the isosbestic point (405 nm) of *p*-nitrophenol and *p*-nitrophenolate and kinetic constants calculated on the basis that the molar absorption coefficient ($\triangle\varepsilon$) was 2104 mol$^{-1}$·cm$^{-1}$. The reaction mixture (1.0 mL) contained 20 μL of a *p*NPH solution (final concentration 25 mM in a mixture of isopropanol and dimethyl sulfoxide at a volume ratio of 3:1) as the substrate, 960 μL of esterase assay buffer (50 mM Tris-HCl, pH 8.0), and 20 μL of the appropriately diluted enzyme. The reaction mixture was incubated at a specific temperature for 5 min, and the reactions were terminated by the addition of 1 mL of ethanol. We evaluated substrate specificity by means of the following substrates: *p*NPA, *p*NPB, *p*NPH, *p*NPC, *p*NPD, *p*NPL, *p*NPM, and *p*NPP. Enzymatic assays were carried out under standard assay conditions.

The effects of pH on the hydrolytic activity of Lx-Est$_{BAS}$ΔSP and free Est$_{BAS}$ΔSP were examined over a pH range from 4 to 10 for 5 min (60 °C). The following buffer systems were utilized: 100 mM citric acid-sodium citrate (pH 4.0–5.0), 200 mM sodium phosphate (pH 6.0–7.0), 50 mM Tris-HCl (pH 8.0), and 50 mM glycine-NaOH (pH 9.0–10.0) [41]. To identify the optimum temperature for enzymatic activity, assays were performed over a temperature range of 20 to 90 °C (pH 8.0). The thermal stability of Est$_{BAS}$ was evaluated by assaying its residual activity after incubation of the enzyme at 60 °C for a specific period. Aliquots were removed at various time points, and residual esterase activity was determined. A control with nonincubated enzyme was used to determine 100% activity. Enzymatic-activity experiments were conducted three times independently.

### 3.7. Effects of Metal Ions, Detergents, and Organic Solvents on Est$_{BAS}$ΔSP Activity

The effects of metal ions (Ca$^{2+}$, Mg$^{2+}$, Cu$^{2+}$, Co$^{2+}$, Ni$^{2+}$, Mn$^{2+}$, Zn$^{2+}$, and Fe$^{2+}$) on the activity of Lx-Est$_{BAS}$ΔSP and free Est$_{BAS}$ΔSP were determined at final concentrations of 1, 5, and 10 mM.

The influence of detergents on the activity of Lx-Est$_{BAS}$ΔSP and of free Est$_{BAS}$ΔSP was analyzed by incubation of the enzyme with a detergent (Tween 20, Tween 60, Tween 80, SDS, or Triton X-100) for 30 min at 4 °C at a final concentration of 1% and 10%.

To estimate the organic-solvent tolerance of Est$_{BAS}$ΔSP, Lx-Est$_{BAS}$ΔSP or free Est$_{BAS}$ΔSP was mixed with various organic solvents at the final concentration of 90% (*v/v*), as described by Li et al. [38].

Lx-Est$_{BAS}$ΔSP or free Est$_{BAS}$ΔSP was mixed with an organic solvent (1, 4-dioxane, ethanol, acetone, or acetonitrile) in sealed vessels and incubated for 70 h (4 °C, 200 rpm). The remaining solvent was removed by centrifugation for 10 min (8000× *g*), and the residual solvent was removed by evaporation. Lx-Est$_{BAS}$ΔSP was resuspended in Tris-HCl buffer (50 mM, pH 8.0), and residual activities were determined under standard conditions.

### 3.8. Characterization of Lx-Est$_{BAS}$ΔSP by Scanning Electron Microscopy (SEM) and FTIR

The surface morphology of Est$_{BAS}$ΔSP before and after immobilization was characterized by SEM and FTIR. SEM images (Supplementary Figure S1) were acquired using an S-3400N scanning electron microscope (Hitachi, Japan). FTIR spectra were obtained on a Nicolet 6700 (Thermo Fisher Scientific, USA) using KBr pellets. FTIR (KBr): $\widetilde{v}$ = 3189, 3063, 3060, 1636, 1631, 908 cm$^{-1}$.

### 3.9. Regioselective Synthesis of Chloramphenicol Esters by Lx-Est$_{BAS}$ΔSP and Free Est$_{BAS}$ΔSP

Chloramphenicol palmitate was synthesized using Lx-Est$_{BAS}$ΔSP or free Est$_{BAS}$ΔSP as the catalyst. The reaction was carried out, and the physical and chemical conditions were varied. A 10-mL round-bottom flask containing a mechanical stirrer bar was filled with acetone, chloramphenicol (0.15 M), vinyl palmitate (0.75 M), and Lx-Est$_{BAS}$ΔSP or free Est$_{BAS}$ΔSP. The reaction temperature was set to 50 °C (24 h, 200 rpm). Subsequently, samples were taken out at equal intervals, and the mixture was centrifuged for 1 min (10,000× *g*). Next, ~0.5 mL of the supernatant was withdrawn for high-performance liquid chromatography analysis.

The conversion ratio of chloramphenicol is represented by the reduction rate of chloramphenicol in the resultant chromatograms:

$$\text{conversion ratio} = 1 - A_t/A_0$$

where $A_0$ is the area of the chloramphenicol peak at time 0, and $A_t$ is the area of the chloramphenicol peak at time *t*.

### 3.10. Reusability of Lx-Est$_{BAS}$ΔSP

After esterification of chloramphenicol with vinyl palmitate, Lx-Est$_{BAS}$ΔSP was recovered and washed 3 times with *n*-hexane (20 mL). The recovered Lx-Est$_{BAS}$ΔSP was dried for 24 h in a desiccator before the next cycle. The activity of the first-cycle reaction was set to 100%, and the activity of the subsequent reactions was calculated based on the first-cycle reaction.

### 3.11. Chloramphenicol Palmitate Characterization

TLC was used to analyze the lipid fractions of the reaction. Lipid fractions (1 μL) in hexane prepared were spotted on the line of origin and run for 30 min in a solvent tank comprising 2 solvents of different ratios i.e., hexane/ ethyl acetate (70:30,*v/v*%) at room temperature. The ZF-1 UV analyzer visualized the spots at 254 nm [42].

The conversion was calculated by high pressure liquid chromatography (Waters 2690 Separations Module; Waters 2487 UV vis Dual Absorbance Detector; readings were made at 273 nm) using a silica gel column (Agilent C18, 4.6 × 150 mm) and eluted with 0.01 M phosphoric acid/acetonitrile (85:15, *v/v*%) with a flow rate of 0.6 mL/min [8]. The products were identified by FTIR, [1]H-NMR (400 MHz, CDCl$_3$), and [13]C-NMR (101 MHz, CDCl$_3$) on a JEOL RESONANCE ECZ 400S spectrometer. CDCl$_3$ was used as a solvent and trimethylsilane was used as an internal reference. Multiple signals of chloramphenicol palmitate are shown in the spectrum of [1]H-NMR and [13]C-NMR (Supplementary Figure S2). FTIR (KBr): $\widetilde{v}$ = 3504, 3323, 2918, 2850, 1742 cm$^{-1}$. [1]H-NMR (CDCl$_3$, 400 MHz): δ 0.87 (*t*, 3H, CH$_3$), 1.26 (s, 24H, CH$_2$), 1.62 (*t*, 2H, CH$_2$), 2.37 (*t*, 2H, CH$_2$), 4.17–4.19 (m, 1H, CH), 4.38–4.48 (m, 2H, CH$_2$), 5.04 (d, 1H, CH), 5.74 (s, 1H, CH), 6.84 (d, 1H, NH), 7.55 (d, 2H, CH), 8.21 (d, 2H, CH). [13]C-NMR (CDCl$_3$, 101 MHz): δ 14.2 (CH$_3$), 22.8 (CH$_2$), 24.8 (CH$_2$), 24.9 (CH$_2$), 29.2–29.8 (8CH$_2$), 32.0

(CH$_2$), 33.8 (CH$_2$), 34.2 (CH$_2$), 54.2 (CH), 62.4(CH$_2$), 66.1(CH), 70.8 (CH), 123.8 (2CH), 126.8 (2CH), 147.1 (C), 147.8 (C), 164.5 (C), 174.6 (C).

## 4. Conclusions

In this research, a novel esterase gene, *est$_{BAS}$*ΔSP, was cloned from *B. altitudinis*. Est$_{BAS}$ΔSP exhibited maximum activity at 50 °C and at pH 8.0. The recombinant protein was successfully immobilized on a novel epoxy resin Lx-105s with a high loading of 96 mg/g, exhibiting enzyme activity over a wider range of pH and temperature. Compared with free Est$_{BAS}$ΔSP, the optimum temperature of Lx-Est$_{BAS}$ΔSP is 60 °C, and the enzyme retained more than 35% activity at 90 °C. In addition, Lx-Est$_{BAS}$ΔSP showed improvements in thermostability and organic-solvent tolerance when compared to free Est$_{BAS}$ΔSP. Moreover, Lx-Est$_{BAS}$ΔSP has excellent catalytic efficiency in the synthesis of chloramphenicol palmitate with regioselectivity. Conversion efficiency reached 94.7% at 0.15 M of substrate after 24 h. Lx-Est$_{BAS}$ΔSP was reused for seven cycles, retaining over 80% of its original activity. In short, this novel esterase holds promise for industrial applications of chloramphenicol palmitate synthesis.

**Supplementary Materials:** The following are available online at http://www.mdpi.com/2073-4344/9/7/620/s1, Figure S1: SEM of the surface of epoxy resin Lx-105s before (a–c) and after (d–f) immobilization of Est$_{BAS}$ΔSP. Magnification: a,d × 100; b,e × 500; c,f × 15 k, Figure S2. (a) $^1$H-NMR spectrum of chloramphenicol palmitate in CDCl$_3$; (b) $^{13}$C-NMR spectrum of chloramphenicol palmitate in CDCl$_3$.

**Author Contributions:** F.D., D.W. and W.W. conceived and designed the experiments; F.D. and X.T. performed the experiments; F.D. and X.Y. analyzed the data; F.D. wrote the paper; L.L. and D.H. contributed reagents/materials/analysis tools; F.D. and W.W. revised the paper.

**Funding:** This research was financially supported by the National Natural Science Foundation of China (No. C31570795), Shanghai Outstanding Technical Leaders Plan 19XD1431800, National Natural Science Foundation of China (Grant No. 81830052, 81530053), and Shanghai Key Laboratory of Molecular Imaging (18DZ2260400).

**Acknowledgments:** Authors are thankful to Xiaoyu Huang and Guolin Lu for providing the precious help of the analysis of NMR spectrum in the present study.

**Conflicts of Interest:** The authors declare no conflict of interest.

## Abbreviations

| | |
|---|---|
| FTIR | Fourier transform infrared spectroscopy |
| IPTG | isopropyl-β-ᴅ-thiogalactopyranoside |
| *p*-NP | p-nitrophenyl |
| *p*NPA | *p*-NP acetate |
| *p*NPB | *p*-NP butyrate |
| *p*NPC | *p*-NP caprylate |
| *p*NPD | *p*-NP decanoate |
| *p*NPH | *p*-NP hexanoate |
| *p*NPL | *p*-NP laurate |
| *p*NPM | *p*-NP myristate |
| *p*NPP | *p*-NP palmitate |
| SDS-PAGE | sodium dodecyl sulfate polyacrylamide gel electrophoresis |
| SEM | scanning electron microscopy; |
| SP | signal peptide; |
| TLC | thin-layer chromatography |

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
