# Peer review of "Immobilization of a Novel ESTBAS Esterase from Bacillus altitudinis onto an Epoxy Resin: Characterization and Regioselective Synthesis of Chloramphenicol Palmitate"

_catalysts, doi:10.3390/catal9070620_

Round 1
Reviewer 1 Report
This article describes the expression of a Bacillus altitudinis esterase in E. coli, its immobilization on epoxy resin and its application to regioselective synthesis of chloramphenicol palmitate. The properties of the immobilized biocatalyst are investigated, and significant amelioration of enzyme stabiity are reported.
A novel esterase with potential applicability to relevant reactions is proposed, and the work presented here could be of interest. A considerable amount of work is described. However, there are a number of aspects that would require improvement before considering publication (including additional experiments):
- the influence of enzyme load on immobilization results should be given in terms of enzyme amount per support amount
- the hypothesis of esterase denaturation with long-term shaking (lines 146-148) should be supported with additional data
- selection of enzyme reaction conditions should be more clearly explained (i.e. 60ºC and pH 8.0, lines 184-185)
- Substrate specificity and the effect of detergents and metal ions should be studied not only for purified recombinant EstBASΔSP, but also for the immobilized enzyme
- The cause of enzyme activity decrease with reuse should be ascertained (lines 299-300). The hypothesis of enzyme leakage can be specifically verified (or discarded)
- Methods descriptions should be improved (i.e. analytical methods should be explained separately)
Therefore, since additional experiments would be required, I recommend the article to be rejected and resubmitted.
Author Response
Responds to Reviewer 1:
Thank you very much for your beneficial suggestion. We highly appreciate your carefulness review and the broad knowledge on the relevant research fields. We are sorry for making some mistakes in the former manuscript. The suggestion are very helpful for writing the revised manuscript. The following are the responses, which we hope to meet with your approval. Thank you.
1. The influence of enzyme load on immobilization results should be given in terms of enzyme amount per support amount
Response: Thank you for giving us this beneficial suggestion. We have added this content in line 5-7, 145-160.
The revised manuscript:
EstBASΔSP was immobilized to a novel epoxy resin (Lx-105s) with a high loading of 96 mg/g.
2. The hypothesis of esterase denaturation with long-term shaking (lines 146-148) should be supported with additional data
Response: Thank you for giving us this beneficial suggestion. Free EstBASΔSP mixed with Lx-105s support was stirred in a thermostatic air bath shaker at 30 °C, and the enzyme activity decreased with the time goes on. In addition, we found that the activity of free EstBASΔSP decreased about 2.1% in 7 hours compared with the initial enzyme activity (data not shown in this manuscripts).
3. Selection of enzyme reaction conditions should be more clearly explained (i.e. 60ºC and pH 8.0, lines 184-185)
Response: Thank you for giving us this beneficial suggestion. In fact, firstly, we studied the effects of pH and temperature on the activity of free EstBASΔSP and Lx- EstBASΔSP (line 202-203), so the optimum temperature and pH of the enzyme are determined. In order to compare the properties of the free enzyme with the immobilized enzyme, this parts are placed in the back part of the article. So other enzyme reaction conditions were at 60 ºC and pH 8.0.
4. Substrate specificity and the effect of detergents and metal ions should be studied not only for purified recombinant EstBASΔSP, but also for the immobilized enzyme.
Response: Thank you for giving us this beneficial suggestion. We had added this part in the manuscript in line 260-271.
The revised manuscript:
Table 1. Effects of metal ions on free EstBASΔSP and Lx-EstBASΔSP activity
Residual | activity (%)a | ||
(1 mM) Fb Lc | (5 mM) Fb Lc | (10 mM) Fb Lc | |
None | 100±0.0 100±0.0 | 100±0.0 100±0.0 | 100 ±0.0 100 ±0.0 |
Mg2+ | 105±2.1 108±1.9 | 103±1.9 104±2.1 | 88.2±1.9 99.8±1.7 |
Zn2+ | 96.1±3.7 97±2.0 | 85.1±2.7 87±2.0 | 68.5±3.5 70±1.2 |
Co2+ | 92.7±2.9 100±1.7 | 81.8±3.5 96±2.5 | 71.4±2.4 74±1.5 |
Ni2+ | 91.5±2.3 92±2.5 | 90.6±2.4 91±2.3 | 83.2±2.0 87±2.2 |
Fe2+ | 92.8±3.4 93±1.8 | 83.4±2.6 84±1.7 | 68.0±1.7 76±1.6 |
a The Free EstBASΔSP and Lx-EstBASΔSP were pre-incubated with the metal ions for 1 h at 4 °C. The presented results are the average of three repeated experiments with ± standard deviation from the mean. The activity toward pNPH without any metal ions was taken as 100%. b F: Free EstBASΔSP. c L: Lx-EstBASΔSP.
Table 2. Effect of detergents on free EstBASΔSP and Lx-EstBASΔSP activity
Residual | Activity (%)a | |
1% Fb Lc | 10% Fb Lc | |
None | 100±0.0 100±0.0 | 100±0.0 100±0.0 |
Tween20 | 94.0±2.1 95±1.9 | 40.5±1.9 41±1.1 |
Tween60 | 94.5±1.6 96±2.1 | 45.6±1.8 47±1.9 |
Tween80 | 94.8±1.8 95±2.3 | 53.3±2.1 55±2.1 |
SDS | 18.9±1.2 20±1.7 | 0 0 |
TritonX-100 | 36.1±3.7 38±2.9 | NDd 8±0.9 |
a Assay was performed under optimum conditions. The residual activity was measured in 50 mM Tris-HCl buffer (pH 8.0) at 60 °C using pNPH as the test substrate. Values represent the mean ± standard deviation (n= 3) relative to the untreated control samples. The activity measured without additives was defined as 100%. b F: Free EstBASΔSP. c L: Lx-EstBASΔSP. d ND: not detectable.
5. The cause of enzyme activity decrease with reuse should be ascertained (lines 299-300). The hypothesis of enzyme leakage can be specifically verified (or discarded)
Response: Thank you for giving us this beneficial suggestion. We have modified the cause of enzyme activity decrease with reuse in line 334-337. As the main aim of this paper is the application of the enzyme, the hypothesis of enzyme leakage has been deleted according to your suggest. Thank you.
The revised manuscript:
However, the decrease of activity was due to the bound water molecules were stripped from the enzyme by hydrophilic organic solvents in the reaction system, the increased hydrophilic solvent content resulted in reduction of enzyme activity.
6. Methods descriptions should be improved (i.e. analytical methods should be explained separately)
Response: Thank you for giving us this beneficial suggestion. We had rewritten this part and analytical methods was explained separately in line 466-472, 494-508.
The revised manuscript:
(Line 466-472)
3.8. Characterization of Lx-EstBASΔSP by scanning electron microscopy (SEM) and FTIR
The surface morphology of EstBASΔSP before and after immobilization was characterized by SEM and FTIR. SEM images (Supplementary Figure S1) were acquired using an S-3400N scanning electron microscope (Hitachi, Japan). FTIR spectra were obtained on a Nicolet 6700 (Thermo Fisher Scientific, USA) using KBr pellets.
(Line 494-508)
3.11. Chloramphenicol Palmitate Characterization
TLC was used to analyze the lipid fractions of the reaction. Lipid fractions (1 μL) in hexane prepared were spotted on the line of origin and run for 30 min in a solvent tank comprising two solvents of different ratios i.e., hexane/ ethyl acetate (70:30, v/v %) at room temperature. The ZF-1 UV analyzer visualized the spots at 254 nm [42].
The conversion was calculated by high pressure liquid chromatography (Waters 2690 Separations Module; Waters 2487 UV vis Dual Absorbance Detector; readings were made at 273 nm) using a silica gel column (Agilent C18, 4.6×150 mm) and eluted with 0.01 M phosphoric acid/acetonitrile (85:15, v/v %) with a flow rate of 0.6 mL/min [8]. The products were identified by FTIR, 1H-NMR (400 MHz, (CD3)2CO), and 13C-NMR (101 MHz, (CD3)2CO) on a JEOL RESONANCE ECZ 400S spectrometer. Deuterated acetone was used as a solvent and trimethylsilane was used as an internal reference. Multiple signals of chloramphenicol and chloramphenicol palmitate are accordingly marked in the spectrum of 1H-NMR and 13C-NMR (Supplementary Figure S2).

Reviewer 2 Report
In the paper entitled "Immobilization of a novel ESTbas esterase from Bacillus altitudinis onto an epoxy resin: Characterization and its regioselective synthesis of chloramphenicol palmitate", the authors have reported results related to an interesting topic, especially from a biotechnological point of view. The experiment were designed very well and also the results are interesting, but ther are points that need to be improved.
Below are reported the critical points that I found:
- page 2, line 39: please specify MRSA;
- page 2, line 62: it is reported the reference Lin et al, but it is not reported in the list of "References";
- page 3, line 105: Figure 1 reported a series of informations but also looking in Mat & Met , are not reported basic information like as: a) What are the proteins used in the alignment of Fig 1A; What is the level of identity among there, and wha program has been used for the alignment? b) What is the strucutre used as template to obtain the 3D model of ESTbas (Fig 1B)? c) Again, what is known about the catalytic activity of the other esterases reported in the alignment of Fig 1A?
- page 5, line 152: although the results reported in paragraph 2.4 are interesting, they are not foundamental in the paper and could be placed as supplemental information. In any case, the physical intarction between enzyme/resin can be evaluated by changing the pH of the buffer in which are suspended the samples;
- page 8, line 220: please indicate the substrate as pNP-C6;
- page 10, line 287: in the sentence "free enzyme ..." seems that the parcentage of conversion are very different, but looking figure 8a, after 3 h the conversion is comprise between 15-20% and after 24 h between 90-95%, the performance are very similar. This is good because the enzyme immibilization do not interfere with catlytic activity;
- page 11, line 308: insert the correct reference of Yoshimoto et al. in the list of "Reference" because the reported ref 38 is not correct;
- page 14, line 409: please define the "molar absorption coefficient" esd for p-nitrophenol;
- page 14, line 418: the dependence of activity versus pH has been mesured at the isosbestic point of p-nitrophenol and p-nitrophenoxide ion?
In conclusion, my opinion is that the manuscript need to a major revision following the above reported points before publication.
Best regards
Author Response
Responds to Reviewer 2:
Thank you very much for your beneficial suggestion. We highly appreciate your carefulness review and the broad knowledge on the relevant research fields. We are sorry for making some mistakes in the former manuscript. The suggestion are very helpful for writing the revised manuscript. The following are the responses, which we hope to meet with your approval. Thank you.
1. Page 2, line 39: please specify MRSA
Response: Thank you for giving us this beneficial suggestion. We have specified MRSA in line 29.
The revised manuscript:
Chloramphenicol was notably efficient against 95% of Gram-negative bacteria and against a variety of Gram-positive aerobic strains (including the dreaded strains of methicillin-resistant Staphylococcus aureus).
2. Page 2, line 62: it is reported the reference Lin et al, but it is not reported in the list of "References";
Response: We are terribly sorry for making this mistake. We mistakenly write Lv as Lin. We have added this reference in the list of "References", the number of this reference is 8.
The revised manuscript:
(8. Lv, D. S.; Xue, X. T.; Wang, N.; Wu, Q.; Lin, X. F., Enzyme catalyzed synthesis of some vinyl drug esters in organic medium. Preparative Biochemistry. 2004, 34, 97-107.)
3. Page 3, line 105: Figure 1 reported a series of informations but also looking in Mat & Met, are not reported basic information like as: a) What are the proteins used in the alignment of Fig 1A; What is the level of identity among there, and what program has been used for the alignment? b) What is the structure used as template to obtain the 3D model of ESTBAS (Fig 1B)? c) Again, what is known about the catalytic activity of the other esterases reported in the alignment of Fig 1A?
Response: Thank you for giving us this beneficial suggestion. We have added these information in the manuscripts.
The revised manuscript:
a) The alignment analysis including alignment protein and the level of identity was in line 95-106, 116-120.
The nucleotide sequence had been deposited in the GenBank under the accession number MH898984. Homology sequence analysis revealed that the EstBAS of B. altitudinis shared the highest identity (99%) with the triacylglycerol lipase of Bacillus sp. and B. pumilus (GI: 639684939, 408690817), and it was 94% identical to the triacylglycerol lipase of Bacillus sp. LLTC93 (GI: 1360561895) (92% to B. pumilus AVI41987; GI: 1351330082, and B. safensis; GI: 408690815 and GI:1130455194, 77% to B. atrophaeus 010787545 and B. atrophaeus 063638677; GI:498485857 and 1027703361, 75% to B. velezensis; GI: 818920927) (Figure 1). Visual inspection of the sequence alignment analysis revealed conservation of amino acids in regions associated with catalysis and stabilization of the protein, e.g., the catalytic triad Ser111, Asp167, and His190 residues (Figure 1a)
b) The structure of EstBAS was generated with a specific template (PDB code: 1T4M, 79.01% identical) in line 107-111, 119-120.
As this esterase had not been previously reported, the secondary and three dimensional structure of EstBAS was predicted by the SWISS-MODEL server and the protein structure of EstBAS was generated with a specific template (PDB code: 1T4M, 79.01% identical) (Figure. 1 b, c), which was viewed by PDB Viewer.
c) By comparing with other esterase reported in the alignment, a conserved domain is found, and then the catalytic center is determined according to the characteristics of the esterase catalytic center.
4. Page 5, line 152: although the results reported in paragraph 2.4 are interesting, they are not fundamental in the paper and could be placed as supplemental information. In any case, the physical intarction between enzyme/resin can be evaluated by changing the pH of the buffer in which are suspended the samples;
Response: Thank you for giving us this beneficial suggestion. We have transferred this part to the supplementary material (Figure S1).
5. Page 8, line 220: please indicate the substrate as pNP-C6;
Response: We are terribly sorry for making this mistake. We have added the reaction substrate p-NP hexanoate (pNPH) (line 276).
The revised manuscript:
Table 3. Effects of organic solvents on free EstBASΔSP and Lx-EstBASΔSP activity
Residual | Activity (%)b | ||
Fb | Lc | ||
Control | 100 ±1.2 | 100±0.9 | |
1,4-dioxane | -1.0 | 12.6±0.7 | 80.2±2.9 |
Ethanol | -0.24 | 52.5±1.8 | 92.5±1.6 |
Acetone | -0.23 | 25.0±1.1 | 93.8±2.6 |
Acetonitrile | -0.15 | NDd | 92.0±1.5 |
alogP value is the partition coefficient of an organic solvent between water and n-octanol phases. After incubating ESTBASΔSP for 70 h in four organic solvents, the residual activity was measured in 50 mM Tris-HCl buffer (pH 8.0) at 60 °C using pNPH as the test substrate. An enzyme sample incubated in buffer only was used as the measure of 100% activity. b F: Free EstBASΔSP. c L: Lx-EstBASΔSP. dND. Not detectable.
6. Page 10, line 287: in the sentence "free enzyme ..." seems that the percentage of conversion are very different, but looking figure 8a, after 3 h the conversion is comprise between 15-20% and after 24 h between 90-95%, the performance are very similar. This is good because the enzyme immibilization do not interfere with catlytic activity;
Response: Thank you very much for your valuable comments. We have modified the sentence (line 317-319).
The revised manuscript:
Free enzyme had a slightly faster conversion rate than immobilized enzyme in the first three hours, then the immobilized enzyme maintained a similar conversion trend with the free enzyme from 6 to 24 h.
7. Page 11, line 308: insert the correct reference of Yoshimoto et al. in the list of "Reference" because the reported ref 38 is not correct;
Response: We are terribly sorry for making this mistake. We have added this reference in the supplementary material due to these contents have been transferred into the supplementary material (Figure S2).
The revised manuscript:
(Yoshimoto, K.; Itatani, Y.; Tsuda, Y., 13C-nuclear magnetic resonance (nmr) spectra of o-acylglucoses. Additivity of shift parameters and its application to structure elucidations. Chemical & Pharmaceutical Bulletin. 1980, 28, 2065-2076.)
8. Page 14, line 409: please define the "molar absorption coefficient" esd for p-nitrophenol;
Response: Thank you for giving us this beneficial suggestion. We had defined the "molar absorption coefficient" in the article (line 431-433).
The revised manuscript:
Reactions were monitored at the isosbestic point (405 nm) of p-nitrophenol and p-nitrophenolate and kinetic constants calculated on the basis that the molar absorption coefficient (△ε) was 2104 M-1·cm-1.
9. Page 14, line 418: the dependence of activity versus pH has been measured at the isosbestic point of p-nitrophenol and p-nitrophenoxide ion?
Response: Thank you for giving us this beneficial suggestion. The chromatometry was the standard and commonly used method to determine the esterase activity. P-nitrophenol and p-nitrophenoxide ion had the same isosbestic point (line 431-433).
The revised manuscript:
Reactions were monitored at the isosbestic point (405 nm) of p-nitrophenol and p-nitrophenolate and kinetic constants calculated on the basis that the molar absorption coefficient (△ε) was 2104 M-1·cm-1.

Reviewer 3 Report
In publication is described application of esterase from Bacillus altitudinis in synthesis of chloramphenicol palmitate. Enzyme was immobilized to a novel epoxy resin (Lx-105s). The authors investigated the influence of many factors on esters synthesis. This publication seems to be within the scope of journal. However it needs several corrections to be more acceptable for publication.
1. In abstract, value of optimal pH should be placed.
2. In introduction, short explanation is necessary, why obtaining of chloramphenicol palmitate is important.
3. In text is necessary explanation, why did the authors optimize the synthesis of p-nitrophenyl hexanoate and not chloramphenicol palmitate.
4. Publication would gain on readability if it has contained a reaction scheme.
5. In manuscript should be placed discussion of obtained result with literature date about other lipases. Short explanation is necessary, why the best activity of enzyme was in case of p-nitrophenyl hexanoate, why only Mg2+ activated enzyme and why addition of all detergents inhibited enzyme.
6. Figure 9 is illegible. Each NMR spectrum should be in separate figure. In my opinion NMR spectrum should be in Supplementary Material but in Materials and Method description of NMR spectrums should be placed. Description should contain multiplicity of signals.
7. In Materials and Methods conversion and yield of chloramphenicol palmitate should be provided.
8. All used shortcut (e.g. p-NP) must be explained in text of manuscript.
9. Whole text of manuscript and tables must be written in the same font size.
Author Response
Responds to Reviewer 3:
Thank you very much for your beneficial suggestion. We highly appreciate your carefulness review and the broad knowledge on the relevant research fields. We are sorry for making some mistakes in the former manuscript. The suggestion are very helpful for writing the revised manuscript. The following are the responses, which we hope to meet with your approval. Thank you.
1. In abstract, value of optimal pH should be placed.
Response: Thank you very much. We have added the optimal pH in line 4-5.
The revised manuscript:
EstBASΔSP exhibited the highest activity towards p-nitrophenyl hexanoate at the optimum temperature and pH (50 ºC, pH 8.0).
2. In introduction, short explanation is necessary, why obtaining of chloramphenicol palmitate is important.
Response: Thank you for giving us this beneficial suggestion. We have added the importance of chloramphenicol palmitate (line 39-41).
The revised manuscript:
Chloramphenicol palmitate is an important chloramphenicol derivatization, which avoids the bitterness of chloramphenicol and can be rapidly hydrolyzed in vivo into the biologically active drug.
3. In text is necessary explanation, why did the authors optimize the synthesis of p-nitrophenyl hexanoate and not chloramphenicol palmitate.
Response: Thank you for giving us this beneficial suggestion. P-nitrophenyl hexanoate was used for the characterization of enzyme activity, chloramphenicol palmitate is the embodiment of enzyme application. In order to study the substrate specificity of EstBASΔSP, a series of substrate was used to obtain the optimum substrate. In text, our aim is mainly to compare the difference in the conversion rate between the free enzyme and immobilized enzyme under the same conditions, so we did not optimize the synthesis of chloramphenicol palmitate.
4. Publication would gain on readability if it has contained a reaction scheme.
Response: Thank you for giving us this beneficial suggestion. We have added the
reaction scheme in the manuscript (scheme 1).
The revised manuscript:
Scheme 1. Preparation of EstBASΔSP immobilization onto epoxy resin Lx-105s and the application for the regioselective synthesis of chloramphenicol palmitate.
5. In manuscript should be placed discussion of obtained result with literature date about other lipases. Short explanation is necessary, why the best activity of enzyme was in case of p-nitrophenyl hexanoate, why only Mg2+ activated enzyme and why addition of all detergents inhibited enzyme.
Response: Thank you very much. We had added a discussion with literature date about other lipases in line 255-259, 330-337.
The revised manuscript:
(1)Line 255-259: The reason was that the formed covalent bond with the support limited the conformational changes of EstBASΔSP, resulting in a more rigid structure of Lx-EstBASΔSP. However, the loss of enzyme activity was due to some specific interaction of the detergents with the enzyme surface.
Line 330-337: Li et al. also observed that SMG1-F278N immobilized on epoxy resin ECR8285 retained 98% of its initial activity after 7 cycles. The above results proved that enzyme interacted with epoxy resin by covalent binding exhibited the advantage of strong and functional groups of support, which made the enzyme a high retention of the catalytic activities. However, the decrease of activity was due to the bound water molecules were stripped from the enzyme by hydrophilic organic solvents in the reaction system, the increased hydrophilic solvent content resulted in reduction of enzyme activity.
(2) Esterase is a kind of enzyme that show the highest activity toward water-soluble or emulsified esters with relatively short fatty acid chains (carbon chain length <10). EstBASΔSP showed the best activity towards p-nitrophenyl hexanoate (p-NPH), which indicated that EstBASΔSP is a “true” esterase that preferentially hydrolyzes short acyl chain substrates.
(3) line 242-246: The activity of EstBASΔSP was decreased in presence of many metal ions. This results indicated that EstBASΔSP activity did not require the presence of those metal ions, and Zn2+ was the most significant inhibition factor. On the other hand, the activity of EstBASΔSP was slightly increased, similarly, Mishra et al. also observed this activation function by Mg2+. He observed that Lecitase® Ultra phospholipase activity had a 1.5-fold activation in the presence of 10 mM Mg2+. He concluded that Mg2+ could activate the enzyme in a lesser extent, and is also able to function without other metal ions (Reference: Mishra. M. K.; Kumaraguru, T.; Sheelu. G., Lipase activity of Lecitase® Ultra: Characterization and applications in enantioselective reactions. Tetrahedron Asymmetry. 2009, 20, 2854-2860.).
(4) Line 257-259: The esterase activity was decreased by addition of all detergents. The reason was that the esterase activity could slightly be stimulated by some detergents in low concentration (0.5%),but some detergents with a higher concentration could inhibit the esterase activity. In the article, the concentration of some detergents are too high for EstBASΔSP.
6. Figure 9 is illegible. Each NMR spectrum should be in separate figure. In my opinion NMR spectrum should be in Supplementary Material but in Materials and Method description of NMR spectrums should be placed. Description should contain multiplicity of signals.
Response: Thank you for giving us this beneficial suggestion. We have transferred NMR spectrum into the supplementary material and each NMR spectrum in a separate figure. Multiple signals of chloramphenicol and chloramphenicol palmitate are accordingly marked in the spectrum of 1H-NMR and 13C-NMR (Supplementary Figure S2).
7. In Materials and Methods conversion and yield of chloramphenicol palmitate should be provided.
Response: Thank you for giving us this beneficial suggestion. We have added yield of chloramphenicol palmitate in materials and methods (line 482-487).
The revised manuscript:
The conversion ratio of chloramphenicol is represented by the reduction rate of chloramphenicol in the HPLC chromatograms:
conversion ratio = 1−At/A0
Where A0 is the area chloramphenicol at time 0 and At is the area chloramphenicol at time t.
8. All used shortcut (e.g. p-NP) must be explained in text of manuscript.
Response: Thank you very much. We have explained all used shortcut in in text of manuscript and in the part of abbreviations used, which was marked in yellow.
9. Whole text of manuscript and tables must be written in the same font size
Response: Thank you very much. The whole text of manuscript and tables were written in the same font size.

Round 2
Reviewer 1 Report
The revised manuscript has been significantly improved and now warrants publication in Catalysts.
Best regards.
Author Response
Thank you so much for your comments.
Reviewer 2 Report
The revised version of the paper contains all the informations that in the first version were missing.
The paper can be accepted for submission in this version.
Best regards
Author Response
Thank you so much for your comments.
Reviewer 3 Report
Although the authors have improved the manuscript, it still contains serious mistakes.
1. There is mistake in structure of chloramphenicol palmitate – lack of ester bond.
2. NMR spectrums in supplementary information should be better quality.
3. Lack of correct description of spectrums in Materials and Methods. Description should be in form as like presented below spectrum of methyl mandelate: Yield: 2:35 g (72%); 1H NMR (300 MHz, CDCl3): δ = 3:50 (s,1H, OH), 3.76 (s, 3H, OCH3), 5.18 (s, 1H, CH), 7.35-7.44 (m, 5H, C6H5); 13C NMR (300 MHz, CDCl3): δ = 53:108, 72.940, 126.639, 128.587, 128.682, 138.208, 174.211. – IR (KBr): n = 3445 (sb), 2952 (s), 1741 (s), 1205 (s), 697 cm-1 (s).

Author Response
Responds to Reviewer 3:
Thank you very much for your beneficial suggestion. We highly appreciate your carefulness review and the broad knowledge on the relevant research fields. We are sorry for making some mistakes in the former manuscript. The suggestion are very helpful for writing the revised manuscript. The following are the responses, which we hope to meet with your approval. Thank you.
1. There is mistake in structure of chloramphenicol palmitate–lack of ester bond.
Response: Thank you very much. We are terribly sorry for making this mistake. We have corrected the structure of the ester in the manuscript (scheme 1).
The revised manuscript:
Scheme 1. Preparation of EstBASΔSP immobilization onto epoxy resin Lx-105s and the application for the regioselective synthesis of chloramphenicol palmitate.
2. NMR spectrums in supplementary information should be better quality.
Response: Thank you very much. We have added the better NMR spectrums into the supplementary information.
| (a) |
The revised manuscript:
Figure S2. (a) 1H-NMR spectrum of chloramphenicol palmitate in CDCl3.
Figure S2. (b) 13C-NMR spectrum of chloramphenicol palmitate in CDCl3.
3. Lack of correct description of spectrums in Materials and Methods. Description should be in form as like presented below spectrum of methyl mandelate: Yield: 2:35 g (72%); 1H NMR (300 MHz, CDCl3): δ = 3:50 (s,1H, OH), 3.76 (s, 3H, OCH3), 5.18 (s, 1H, CH), 7.35-7.44 (m, 5H, C6H5); 13C NMR (300 MHz, CDCl3): δ = 53:108, 72.940, 126.639, 128.587, 128.682, 138.208, 174.211. – IR (KBr): n = 3445 (sb), 2952 (s), 1741 (s), 1205 (s), 697 cm-1 (s)
Response: Thank you very much. We have added the correct description of spectrums in Materials and Methods in line 479, 515-521.
The revised manuscript:
Line 479
FTIR (KBr): ṽ=3189, 3063, 3060, 1636, 1631, 908 cm-1.
Line 515-521
FTIR (KBr): ṽ=3504, 3323, 2918, 2850, 1742 cm-1. 1H-NMR (CDCl3, 400 MHz): δ 0.87 (t, 3H, CH3), 1.26 (s, 24H, CH2), 1.62 (t, 2H, CH2), 2.37 (t, 2H, CH2), 4.17-4.19 (m, 1H, CH), 4.38-4.48 (m, 2H, CH), 5.04 (d, 1H, CH), 5.74 (s, 1H, CH), 6.84 (d, 1H, CH), 7.55 (d, 2H, CH), 8.21 (d, 2H, CH). 13C-NMR (CDCl3, 101 MHz): δ 14.2 (CH3), 22.8 (CH2), 24.8 (CH2), 24.9 (CH2), 29.2-29.8 (8CH2), 32.0 (CH2), 33.8 (CH2), 34.2 (CH2), 54.2 (CH), 62.4(CH2), 66.1(CH), 70.8 (CH), 123.8 (2CH), 126.8 (2CH), 147.1 (C), 147.8 (C), 164.5 (C), 174.6 (C).
